# Early evolution of the ozone mini-hole generated by the Australian bushfires 2019-2020 observed from satellite and ground-based instruments

Redha Belhadji[1], Pasquale Sellitto[1,2], Maxim Eremenko[1], Silvia Bucci[3], Tran M. Nguyet[1], Martin Schwell[1], Bernard Legras[4].

[1]Univ. Paris Est Créteil and Université de Paris, CNRS, Laboratoire Interuniversitaire des Systèmes Atmosphériques, Institut Pierre Simon Laplace, Créteil, France
[2]Istituto Nazionale di Geofisica e Vulcanologia, Osservatorio Etneo, Catania, Italy
[3]Institut für Meteorologie und Geophysik, University of Vienna, Austria
[4]Laboratoire de Météorologie Dynamique, UMR CNRS 8539, École Normale Supérieure, PSL Research University, École Polytechnique, Sorbonne Universités, École des Ponts PARISTECH, Institut Pierre Simon Laplace, Paris, France

*Correspondence to*: Redha Belhadji (redha.belhadji@gmail.com); Pasquale Sellitto (pasquale.sellitto@lisa.ipsl.fr)

## Abstract

The intense wildfires in Australia, during the 2019-2020 fire season, generated massive Pyro-cumulonimbus (pyro-Cb) clouds, and injected an unprecedented amount of smoke aerosols into the upper troposphere–lower stratosphere (UTLS). The smoke aerosols produced a self-sustaining confined anticyclonic vortex, that ascended up to 35 km altitude by March 2020 by diabatic heating of radiation absorbing aerosols. This vortex transported ozone-poor tropospheric air into the ozone-rich stratosphere, thus forming a transient ozone mini-hole. This study investigates the early evolution of the dynamically-generated ozone mini-hole, using satellite and ground-based observations, supported by modelling information. Ozone anomalies within the vortex are tracked and quantified by satellite observation. In particular, ad-hoc in-vortex observations are derived by coupling the IASI (Infrared Atmospheric Sounding Interferometer) satellite observations and meteorological reanalysis information of the vortex. With these observations, a 30–40% ozone reduction is observed in a 6-km partial stratospheric column, which exponentially decreased to ~7% by the end of January with an e-folding time of about one week, as the vortex ascended in the stratosphere. A total ozone column reduction of ~7%, immediately after the pyro-Cb injection, was observed with IASI and the TROPOMI (TROPOspheric Monitoring Instrument) satellite instrument. Consistently, ground-based measurement at Lauder, New Zealand showed a localised ozone reduction reaching ~10% (total column) and ~20% (in-vortex stratospheric partial column) associated with two vortex overpasses. These results provide insights into the impacts of extreme wildfires and pyro-Cbs on the dynamics and composition of the stratosphere.

## 1. Introduction

Due to their role in the regeneration of forests at the global scale (e.g. (Johnson and Miyanishi, 2001), forest fires are part of the natural cycle of the biosphere. Due to recent human-induced climate change and the increase of favorable conditions for them to develop, the risk of particularly frequent and intense wildfire events has increased (Cunningham et al., 2024). This increase in wildfires frequency and intensity is particularly critical in sensitive areas like Australia (Dowdy et al., 2019). At the turn of the year 2020, southeastern Australia was the theatre of a new record-setting fire event, in terms of its atmospheric impacts. The Australian fire season in 2019-2020, named the "Black Summer", started on September 2019 in New South Wales and Queensland and persisted until the end of February 2020 (Davey and Sarre, 2020). During this period, about 7.4 million hectares of temperate forests across Australia were turned to ash (Peterson et al. 2021). The peak fire activity phase was registered at the end of December 2019 and beginning January 2020, when the intense power of the fires generated massive pyro-cumulonimbus (pyroCb) clouds in the upper-troposphere—lower-stratosphere (UTLS) (Peterson et al., 2021). The observed perturbation in stratospheric composition was unprecedented, in its magnitude, for a fire event (e.g. Ohneiser et al., 2022; Solomon et al., 2022). The injection of gaseous and particulate carbon species was estimated to be 3 time larger than the 2017 Pacific Northwest wildfire event (Yu et al., 2021), which set the previous larger observed impact on stratospheric aerosol load for a wildfire. The stratospheric injection of smoke aerosols was estimated between $0.4 \pm 0.2$ Tg (Khaykin et al., 2020) and $2.1 \pm 1.0$ Tg (Hirsch and Koren, 2021), a mass comparable to the stratospheric aerosol perturbation of a moderate volcanic eruptions (e.g. Kloss et al., 2021). This injection of smoke aerosols produced the strongest documented large-scale radiative forcing for a fire event, of comparable magnitude with the strongest volcanic eruptions of the post-Pinatubo era (Sellitto et al., 2022). In addition, the absorption of solar radiation diabatically heated these large amounts of injected absorbing aerosols (Sellitto et al., 2023) and generated an anticyclonic self-maintained ellipsoidal vortex of 1000 km in diameter (Podglajen et al., 2024), who travelled around the Southern Hemisphere and ascended up to 35 km altitude by March 2020 (Kablick et al., 2020; Khaykin et al., 2020). In his way up to the deep stratosphere, the anticyclonic vortex dragged large amounts of ozone-poor tropospheric air masses who temporarily diluted the ozone-rich stratospheric air and generated an ozone mini-hole (Bernath et al., 2022; Khaykin et al., 2020; Ohneiser et al., 2022; Solomon et al., 2022; Yu et al., 2021). At larger spatiotemporal scales, chemical effects of interaction of the injected aerosol, water vapour and other pollutants, on the stratospheric ozone layer, took place (Ansmann et al., 2022; Solomon et al., 2022).

In this paper we discuss the evolution of the early dynamically-generated ozone mini-hole using multiple satellite observation supported by modelling information. Dedicated in-vortex column estimations of the ozone reduction, at this time scale, where also performed with ad-hoc satellite observations with the Infrared Atmospheric Sounding Interferometer (IASI). With these analyses, we could track the transient ozone reduction as it moves around the Southern Hemisphere and as it ascended in the stratosphere. Ground-based remote sensing observations from the Fourier transform spectrometer based in Lauder (New Zealand) are also used to characterise this ozone mini-hole from a fixed-surface perspective.

This paper is structured as follows. In Sect. 2, the data and methods used in this work are presented. A description of the main vertical structure, associated with the ozone mini-hole, and its dynamical patterns, is given in Sect. 3. Results are shown and discussed in Sect. 4. Conclusions are drawn in Sect. 5.

## 2. Data and methods

### 2.1. Observations

#### 2.1.1. Ozone total and partial column observations with the IASI (Infrared Atmospheric Sounding Interferometer)

The IASI instrument (Clerbaux et al., 2007) is an operational meteorological satellite sensor, operating in the infrared spectral region. Three IASI instruments fly onboard the MetOp-A-C satellite series in a polar sun-synchronous orbit (about 800 km altitude). Each of them samples the globe about twice per day with a nadir geometry. The IASI measures the thermal infrared radiation emitted by the Earth's surface and the atmosphere. It is a Fourier transform spectrometer with a 2 cm optical path difference covering the 645–2760 $cm^{-1}$ spectral range. Its apodised spectral resolution is 0.5 $cm^{-1}$, in terms of its full-width at half-maximum. The radiometric accuracy in noise-equivalent radiance temperature at 280K ranges between 0.28 K at ~650 $cm^{-1}$ and 0.47 K at 2400 $cm^{-1}$ (Eremenko et al., 2008). The horizontal resolution of IASI is 12 km at the sub-satellite point, and one swath of IASI covers about 2200 km in the across-track. In addition to temperature and humidity profiles and cloud information, which are primary goals of the IASI mission, it provides retrievals of different trace gases (e.g. Clarisse et al., 2011) and aerosols (e.g. Clarisse et al., 2013), including their speciation (Guermazi et al., 2021). Height-resolved ozone distributions are retrieved by IASI with different retrieval algorithms. In this work, we use the product developed at LISA (Laboratoire Interuniversitaire des Systèmes Atmosphériques) and described by Eremenko et al. (2008). The LISA retrievals are based on the radiative transfer model KOPRA (Karlsruhe Optimised and Precise Radiative transfer Algorithm, Stiller et al., 2000) and its inversion module KOPRAFIT. The LISA ozone retrieval scheme provides ozone profiles at 1-km vertical sampling and with 3.0-3.5 degrees of freedom for the whole atmospheric column and about 2.0 in the stratosphere (Dufour et al., 2012). The LISA ozone retrieval schema has extensively been validated, e.g. comparing its retrievals with ozone sonde data (Dufour et al., 2012). In this work, in-vortex partial ozone columns in the stratosphere are calculated, based on the LISA ozone vertical profiles retrievals, as described in Sect. 4.1.

#### 2.1.2 Ozone total column observation with the TROPOMI (TROPOspheric Monitoring Instrument)

The TROPOMI (TROPOspheric Monitoring Instrument) instrument, onboard the Copernicus Sentinel-5 Precursor satellite since August 2019, is a nadir-viewing imaging spectrometer that measures the spectral solar radiance scattered at nadir by the Earth. It covers wavelength bands going from the ultraviolet to the shortwave infrared (270–2385 nm) (Veefkind et al., 2012). The instrument was jointly developed by the Netherlands Space Office and the European Space Agency. It provides a quasi-

daily global coverage, with a horizontal resolution of 5.5 km x 3.5 km and a 2600 km swath across-track. For this work the TROPOMI Level 2 Ozone Total Column product (Garane et al., 2019) is used in its v2 version.

### 2.1.3 Ozone vertical profiles observations with the OMPS-LP (Ozone Mapping and Profiler Suite Limb Profiler)

The OMPS (Ozone Mapping and Profiler Suite) suite has been operational aboard the Suomi-NPP (National Polar-orbiting Partnership) satellite, as part of the Joint Polar Satellite System Program (JPSS) (Goldberg and Zhou, 2017), since January 2012. The OMPS suite includes a Limb Profiler (OMPS-LP) that measures vertical profiles of scattered solar radiation in the spectral range 290–1000 nm, using a limb-viewing geometry. The instrument is primarily designed to provide ozone concentration profiles up to 60 km and aerosol extinction profiles up to 40 km altitude, with a vertical resolution of 1 km.

(Taha et al., 2020). The OMPS-LP instrument has three detectors, but the central one provides the highest-quality measurements and is used in this work. For this work, the University of Saskatchewan (USask) OMPS-LP 2D v1.3.0 ozone concentration profiles dataset has been used. These data are based on a tomographic retrieval algorithm which directly accounts for atmospheric variations in the along-orbital-track dimension (Zawada et al., 2018). The retrievals are provided with a vertical sampling of 1 km and a horizontal sampling of approximately 125 km.

### 2.1.4 Ground-based ozone observation with NDACC (Network for the Detection of Atmospheric Composition Change) FTIR (Fourier Transform InfraRed) in Lauder, New Zealand

The Network for the Detection of Atmospheric Composition Change (NDACC) is an international network of ground-based, remote-sensing research stations. Its main objective is to establish long-term observational data sets to monitor atmospheric composition in terms of different atmospheric species, their trends and impacts. The NDACC network includes more than 100

stations, with more than 160 active instruments. Among these instruments are continuously calibrated and validated standardised solar-viewing Fourier Transform InfraRed (FTIR) spectrometers. These instruments record mid-infrared solar transmission spectra at high spectral resolution (0.002 cm$^{-1}$). These spectra contain the signatures of molecular rotational-vibrational transitions of numerous trace gases (including ozone) in the Earth's atmosphere. The spectra are analyzed to measure the concentration of these trace gases in the atmosphere using the pressure and temperature dependence of the

absorption line shapes to retrieve vertical profile information from which ozone vertical concentration profiles are calculated. The retrieval uses the inversion method developed by Rodgers (2000) as implemented by Björklund et al. (2023). The retrieved ozone profiles have generally 3-5 degrees of freedom in the whole atmospheric column. The FTIR spectrometer in the Lauder station, New Zealand (45.08°S, 169.78°E, 370m above sea level, a Bruker 120M model) is used in this work because of its ideal location to observe the overpass of the Australian fires-induced vortex in early 2020, starting from the pyro-convective

injection events and during its first stages of atmospheric dispersion.

**2.2 Reanalysis data: Vorticity and ozone observation with ECMWF (European Centre for Medium-Range Weather)**

The ECMWF (European Centre for Medium-Range Weather) IFS (Integrated Forecasting System) is the operational configuration of the ECMWF global numerical weather prediction system (version CY46R1, https://www.ecmwf.int/en/publications/ifs-documentation). It is based on an atmosphere-land-wave-ocean forecast model and an analysis system that provides an accurate estimate of the initial state of the model. The forecast model has a 9 km horizontal resolution grid and 137 vertical levels, with a top around 80 km altitude. The analysis is based on a 4-dimensional variational method, run twice daily using more than 25 million observations per cycle, primarily from satellites. Ozone observations from TROPOMI are assimilated in ECMWF-IFS reanalyses but not OMPS and IASI (IASI radiances are assimilated). The IFS produces high-resolution operational 10-day forecasts twice daily. The IFS data are used in this work to track the spatiotemporal evolution of the main vortex emanating from the Australian fires 2019-2020 with a high temporal resolution. Most notably, the IFS data are used here to track the vertical ascent of the vortex and is coupled with the IASI ozone vertical profile observations to produce ah-hoc ozone partial columns, which are representative of the in-vortex conditions.

**3. The Australian fires anti-cyclonic vortex**

A remarkable dynamical structure was observed emanating from the main smoke patch that originated from the Australian fires PyroCb event at the turn of the year 2020, as the core plume began to encapsulate into compact bubble-like structures. Different vortices formed, starting on 31st December 2019. The largest isolated structure, measuring about 1000 km in width and 6 km in depth, was first observed with CALIOP (Cloud-Aerosol Lidar with Orthogonal Polarization) space-LiDAR observations on 4th January 2020, in terms of a signature of elevated aerosol extinction (Khaykin et al., 2020), then tracked along its trajectory around the Southern Hemisphere using ECMWF-IFS reanalyses, see Figure 1. The vortex is notably identified by an isolated patch of large vorticity in the stratosphere (Fig. 1a), indicating the transport of tropospheric air masses. This smoke bubble first travelled across the Pacific and then hung above the tip of South America for a week before heading west and crossing zonally the whole Southern Hemisphere at about 30-60°S (Ohneiser et al., 2022). This compact vortex was filled with shortwave-absorbing black carbon aerosol, possibly of a small mean size, which provided a significant localised radiative heating, with heating rates averaging at $8.4\pm6.1$ K d$^{-1}$ in the whole vortex, and with values locally as large as 15-20 K d$^{-1}$ (Sellitto et al., 2023). Due to this diabatic heating, the vortex progressively lofted from the pyro-convective injection heigh (about 17 km) to about 35 km, as it travelled around the Southern Hemisphere (Khaykin et al., 2020). The ECMWF-IFS data also show that this compact vortex was accompanied by a co-localised air volume with negative ozone concentration anomalies in the stratosphere (Fig. 1b), most likely associated with the transport of ozone-poor tropospheric air masses to the ozone-rich stratosphere.

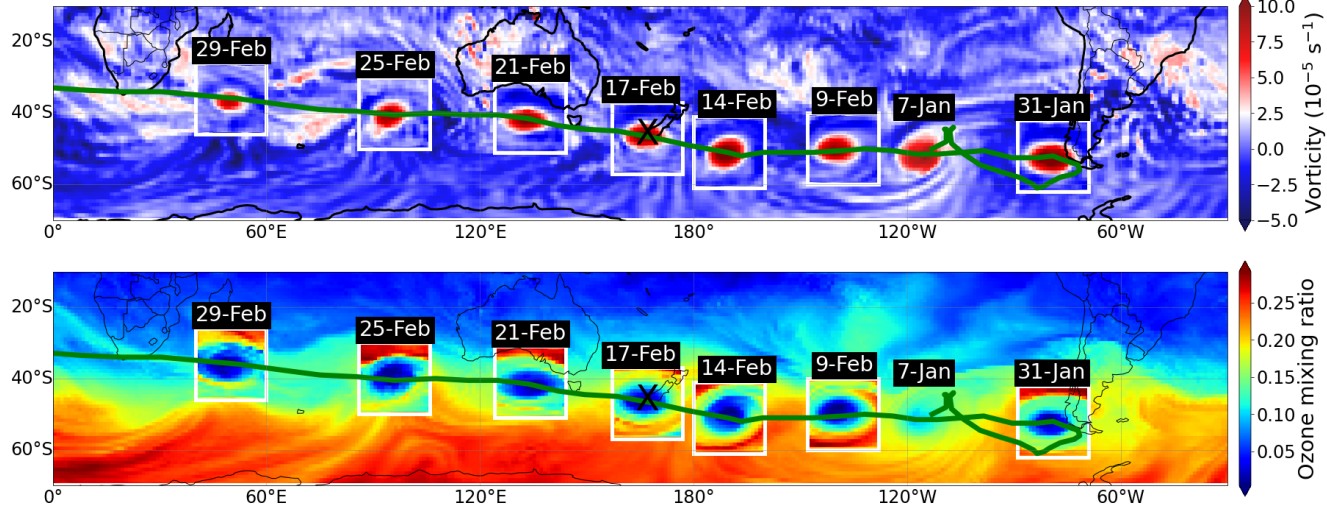


**Figure 1: Horizontal evolution of the vortex shown with relative vorticity (a) and ozone mixing ratio (b), data from ECMWF-IFS reanalyses. The background shows the relative vorticity and ozone mixing ratio on 07 January 2020 18UTC, at 75.2 hPa (~18 km altitude, thus at the vertical location of the vortex) corresponding to the level of highest vorticity and lowest concentration of ozone in the vortex. The boxes show the vorticity field and the ozone anomaly at other indicated times as horizontal sections at the vertical**

**level of the vortex centroid projected onto the background field. The green curve is the daily-sampled trajectory of the vortex centroid. The black cross represents the position of the NDACC FTIR observatory of Lauder New Zealand.**

## 4. Results

### 4.1. The ozone mini-hole evolution from a satellite perspective

We first visualise the vertical evolution of the ozone mini-hole collocated with the Australian fires vortex, during its first

phases of atmospheric evolution, using satellite data

We focus here on the time period immediately after the first pyro-convective injection phase between 31st December 2019 and 4th January 2020 (Peterson et al. 2021). We estimated ozone anomaly profiles using OMPS-LP ozone concentration profile observations. Selected OMPS-LP ozone anomaly observations are shown in Fig. 2 (left column), depending on the availability of OMPS-LP overpasses at the location of the vortex from ECMWF-IFS reanalyses. The ozone anomaly is estimated with

respect to monthly average OMPS-LP ozone concentration profiles (an average of January and February 2020). The first trackable observation of the ozone mini-hole, from OMPS-LP anomalies, was on the 31[st] December 2019, which was the day of one of the main pyro-convective outbreak events (Khaykin et al., 2020). The main vortex is here discussed more in details. For this, an ozone mini-hole is clearly visible by the 7[th] January (Fig. 2), when we observe a clear ozone anomaly around 50-60°S at 17-18 km altitude, which approximately corresponds to the pyro-convection altitude of injection observed by CALIOP

(Sellitto et al., 2023). This is also observed at the horizontal scale at the altitude of the vortex identified by ECMWF-IFS reanalyses (Fig. 2). An interesting feature immediately after the injection is the co-existence of a wavy vertical pattern in the ozone anomaly, with a maximum negative anomaly at about 25 km altitudes and a maximum at about 30 km. This hat-like

shape extended from about 35 to 70°S. From that point, different behaviours are observed. By the 13th January the two anomaly structures seem propagating at slightly different directions, and eventually merging by the 22th January in a hat/mushroom

shape. While the lower-altitude structure is easily identified in the core of the moving vortex, the nature of the higher-altitude ozone anomaly is yet to be determined. The ozone anomaly has been tracked for over a month (see Fig. 2, for selected days in time interval) and is lofted of several km in one month, consistently with ECMWF-IFS reanalyses.

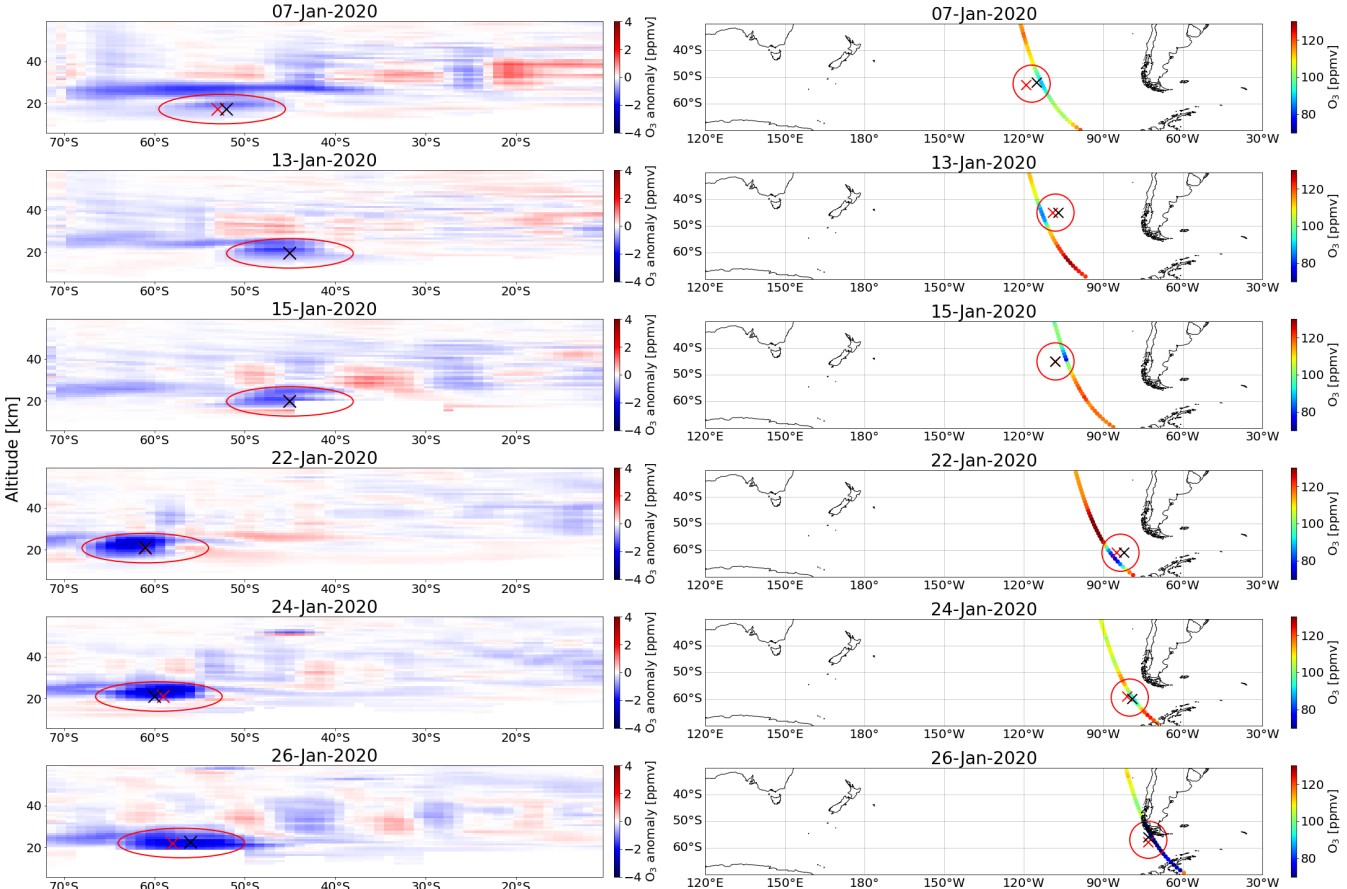

**Figure 2: OMPS-LP ozone anomalies for selected days following the Australian fires, with a latitude/altitude plot (left column).**
**OMPS-LP ozone observation concentration at the core vortex altitude from ECMWF IFS (right column). The circle represents the rough shape of the vortex (~1000 km diameter, in the panels). The red and black crosses show the center of the vortex according to the ECMWF-IFS reanalyses at 6UTC and 18UTC, respectively.**

The horizontal trajectory of the ozone mini-hole is tracked using TROPOMI total ozone column observations. These observations provide a dense horizontal resolution with a very good spatial coverage, thus this is a very effective tool for

tracking relatively small-scale patterns of chemical tracers, like this ozone mini-hole phenomenology. Selected days are shown for the TROPOMI total ozone columns in Fig. 3. The ozone mini-hole signature associated with the vortex, while occurring at a limited vertical interval (about 6 km altitude range in the stratosphere, depending on the lofting stage), is clearly visible in the total column observations during the first month since the initial injection. Comparing with a nearby background (i.e.

outside from vortex area, see also figure 3), with the method extensively discussed later on for IASI observations, TROPOMI

showed a maximum ~8% total ozone reduction due to the ozone mini-hole (case of 7th January 2020 in Fig 3).

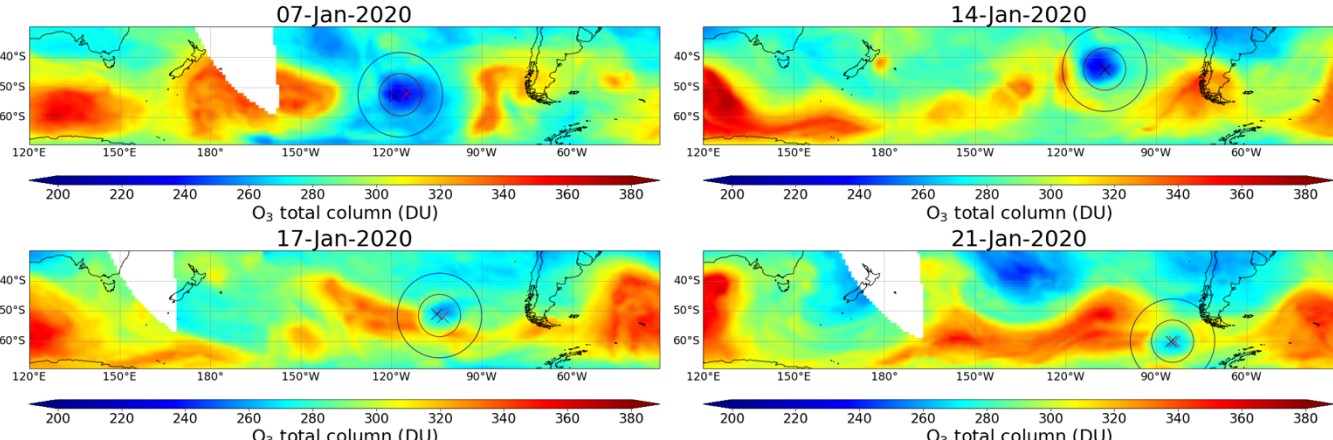

**Figure 3: TROPOMI total ozone column observations, for four selected days in January 2020 (see labels over the individual panels). The inner and outer circles show the averaged regions for the total ozone columns in the case of the in-vortex ozone mini-hole and the background, respectively. The red and black crosses show the center of the vortex according to the ECMWF-IFS**

**reanalyses at 6UTC and 18UTC, respectively.**

While the TROPOMI data provide a spatially-resolved view of the ozone mini-hole, the lack of vertical-specific information prevents us to give quantitative estimation of the Australian-fires-induced ozone reduction associated with the main anticyclonic vortex, with this data set. Vertical profiles observations, at a relatively high horizontal resolution are needed if we want to quantify this effect. The high horizontal resolution requirement excludes OMPS-LP for this estimation. Then, we use

the nadir-looking sounder IASI, which is very well adapted to reach this target. The progressively lofting nature of the vortex and the associated ozone mini-hole, due to diabatic heating, requires a specific vertical tracking, in addition to the horizontal tracking discussed above. Thus, an ad-hoc approach is here used to follow the vertically-localised effect of the vortex on the stratospheric ozone. Using IASI measurements, with LISA ozone retrieval scheme, we have calculated an "in-vortex column", i.e. an ozone column of 6 km depth (which is the approximate depth of the vortex, in this phase), 3km above and 3km under

the center of the vortex obtained with ECMWF-IFS reanalysis data. We have then spatially-averaged these columns in a circular region of 7° radius (empirically selected based on visual observation of the mini-hole signature) in both latitude and longitude around the vortex core, again taken from ECMWF-IFS reanalyses. This method allows the vertical and horizontal tracking of the ozone mini-hole through the stratosphere as the vortex progressively ascended and moves horizontally. Anomalies are then calculated with respect to a "spatial" background of ozone concentrations. This is obtained, for each IASI

observation, with a column of 6 km depth (as for the vortex), in an annular region with an outer radius of 14° in latitude and longitude and removing the 7° radius region of the vortex. The IASI partial stratospheric ozone column, as well as the locations of the vortex and of the spatial background, based on the vortex altitude range defined with the method described above, are shown in the whole region of interest in Fig. 4, for selected days.

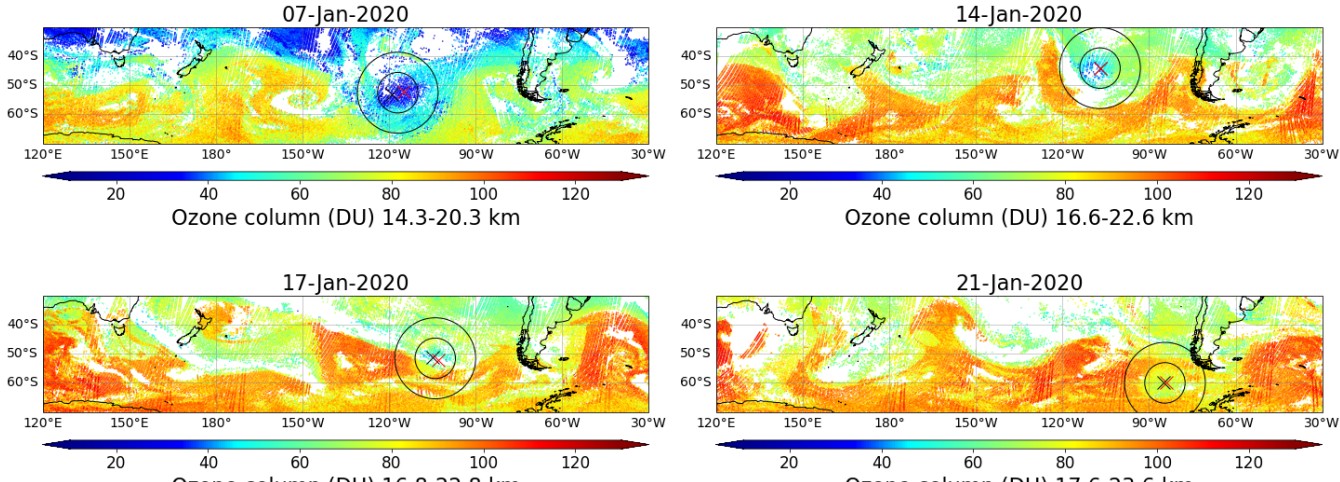

**Figure 4: IASI partial stratospheric ozone column, defined using the vortex altitude derived with ECMWF-IFS reanalyses, for four selected days in January 2020 (see labels over the individual panels). The depth of the partial stratospheric ozone column is 6 km, located at 14.3-20.3 km, 16.6-22.6 km, 16.8-22.8 km and 17.6-23.6 km, for the different days in the panels. The inner and outer circles show the averaged regions for the total ozone columns in the case of the in-vortex ozone mini-hole and the background, respectively. The red and black crosses show the center of the vortex according to the ECMWF-IFS reanalyses at 6UTC and 18UTC, respectively.**

To test the consistency of this approach, we also defined a second approach, in particular defining a different type of background. The in-vortex column, associated with the ozone mini-hole (same as for the previous spatial approach), is compared to a "temporal" background, i.e. the decadal average, from 2010 to 2019, in the same region as the in-vortex column. This method is meant to estimate the ozone reduction, due to the Australian fires vortex in 2020, with respect to climatological values of the stratospheric ozone at the vortex vertical and horizontal position. This contrasts with the spatial background method described before, which was more associated with instantaneous stratospheric ozone perturbations. The IASI stratospheric ozone anomaly, as well as the locations of the vortex and of the spatial background, resulting from the temporal background approach are shown in the whole region of interest in Fig. 5, for selected days. This varies from the spatial approach of Fig. 4 due to the binning in regular pixels, because of the non-identical IASI orbits in same days of different years.

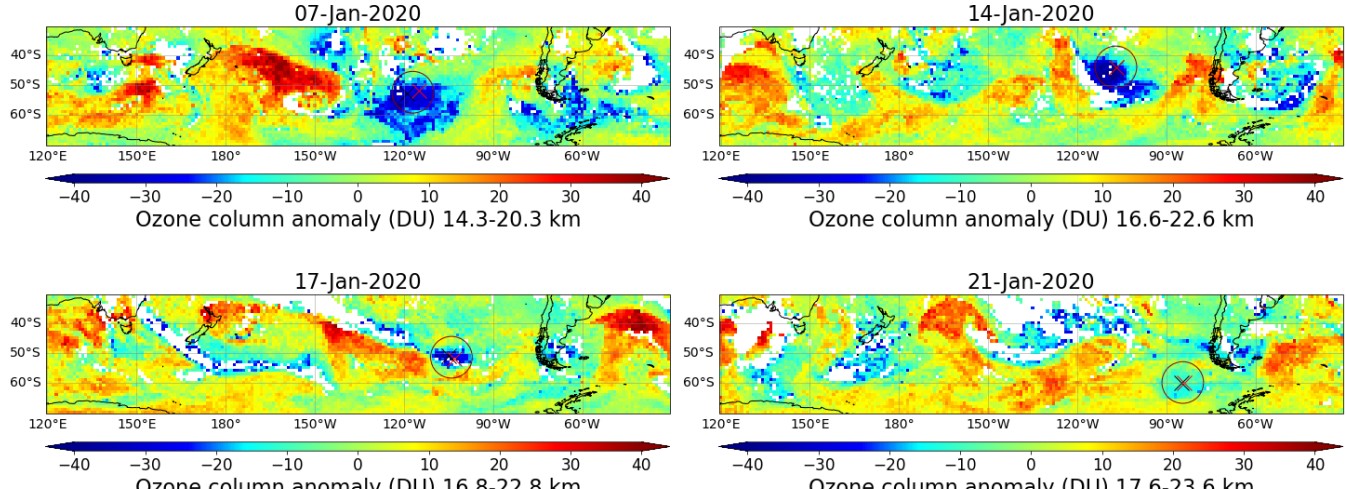


**Figure 5: IASI partial stratospheric ozone column anomaly, based on the temporal background approach, defined using the vortex altitude derived with ECMWF-IFS reanalyses, for four selected days in January 2020 (see labels over the individual panels). The depth of the partial stratospheric ozone column is 6 km, located at 14.3-20.3 km, 16.6-22.6 km, 16.8-22.8 km and 17.6-23.6 km, for the different days in the panels. The anomaly in 2020 is estimated with respect to 2010-2019 averages. The red circles show the averaged regions for the total ozone columns within the in-vortex ozone mini-hole. The red and black crosses show the center of the vortex according to the ECMWF IFS reanalyses at 6UTC and 18UTC, respectively.**


With both spatial and temporal background approaches, the presence of an ozone mini-hole, colocalised with the Australian fires' vortex, is clearly visible for approximately the first month after the pyro-convective injection. It is worth noticing that, visually, the IASI observations of the ozone mini-hole are quite consistent with the geolocalisation of the vortex centroid,

provided by the ECMWF-IFS reanalyses. To display the evolution of the ozone reduction within the mini-hole through its dispersion, time series have been produced using IASI observation data. The percent reduction derived with both the spatial and temporal background methods are shown in Fig. 6. Data for 11st-13rd January 2020 have been excluded because of the lack of a sufficient number of IASI observation in the vortex area during these days. The first period from the 4th to the 6th of January correspond to the phase of separation of the vortex from the main plume and the start of its ascension through the

stratosphere. For these days, the estimation of the in-vortex reduction is difficult due to the noisy background associated with the large-scale aerosol plume, which interacts with the IASI ozone retrievals (i.e. Dufour et al., 2018). The vortex is reasonably separated from the main hemispheric plume on 7th January. Thus, we start our quantitative analyses from this day. An ozone reduction of about 30-40% in the 6-km-deep partial stratospheric column, in the in-vortex ozone mini-hole with respect to the background ozone, is observed on 7th January. Starting from this phase, the reduction follows approximately an exponential

decay temporal trend with an e-folding time of about a week, until reaching about 7% at the end of January 2020. This general behaviour and the quantitative results on the initial reduction and of the e-folding time are found very consistently with the spatial and temporal background methods, which corroborates our interpretations. The observed decrease of the ozone reduction in the vortex follows an exponential-decay law curve possibly due to the almost instantaneous pyro-convective injection of ozone-poor air coming from the troposphere to the stratosphere, and the progressive dilution of the vortex in

ambient air, approaching background levels by the end of the month. By the way, the curve reaches a ~7% minimum which indicates that the local stratospheric ozone amount did not fully reach background levels after one month since the event. This non-zero minimum can be related to partial vortex confinement or to the existence of other processes, like chemical ozone depletion by aerosols or other injected gaseous pollutants in the vortex, further discussed by Hirsch et al. (2021) and Solomon et al. (2022).

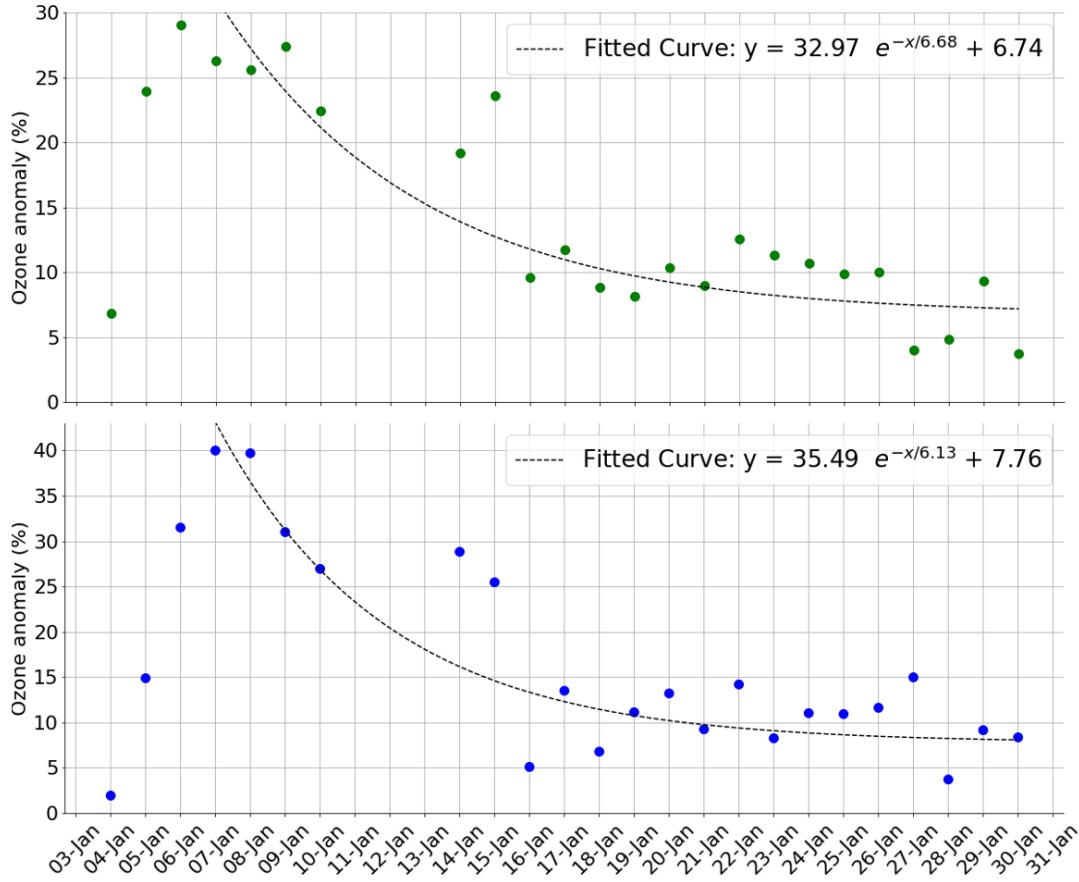


**Figure 6: Temporal evolution of the partial stratospheric ozone column anomaly (percent reduction) for the in-vortex ozone mini-hole observed by IASI, using the spatial background method (see Fig. 4) (a) and using the temporal background method (see Fig. 5) (b). An exponential-decay fit is applied to the two time series and the fitted equation is in the respective panels (y is the ozone percent anomaly and x is time in days).**

As expected, the in-vortex localised ozone reduction in the partial stratospheric ozone columns, observed by IASI and shown in Figs. 4-6, are larger than the total ozone column reduction, observed by TROPOMI and shown in Fig. 3. As a point of comparison, for 7th January the IASI-derived stratospheric ozone reduction is 30-40% with respect to background, while it is ~8% for the TROPOMI total column. It is worth mentioning that the IASI total ozone column reduction for the 7th January was found ~7.5%, which is remarkably consistent with TROPOMI total column results.

## 4.2. The ozone mini-hole evolution from a ground-based perspective

At a longer time-scale than what discussed in Sect. 4.1, the ozone mini-hole was observed by a FTIR spectrometer at the NDACC ground station in Lauder, New Zealand. The Lauder observatory happened to be on the path of the Australian fire vortices in January 2020, as they moved eastwards immediately after the pyro-convective injection, and in February 2020, as they come back westwards in a second phase of dispersion, see Fig. 1. The time series of the Lauder FTIR ozone observations is shown in Fig. 7 and quantitative estimations of the local ozone reduction are in Tab. 1 and 2 (for the overpass of January and February, respectively). During the overpass of the fire plume in its way eastwards, a sharp reduction of the local total in-vortex stratospheric and local ozone concentration, reaching about 12% and 23%, respectively, with respect to background (1st–10th January average), can be observed at Lauder between the 10th and 15th of January (see Tab. 1). These results agree reasonably well with the 20%-25% reduction observed by ozone sonde over Antarctica and New Zealand by Ohneiser et al. (2022) and can possibly be attributed to a secondary vortex (but not to the main vortex shown in Fig. 1). This local reduction was followed by a slow restoration of the ozone, as the vortex move away from Lauder towards South America (Fig. 1). A second overpass at Lauder occurred on the way back of the main vortex (Fig. 1, see in particular the position of Lauder on 17th February 2020, in Fig. 1) and, correspondingly, a second ozone reduction phase was observed by the FTIR in Lauder, reaching about 13% (total ozone column) and 21% (in-vortex stratospheric column) with respect to background (5th-10th February average). The maximum of the ozone reduction is observed on 17th of February, which is very consistent with the core overpass of the moving main vortex shown by ECMWF-IFS data.

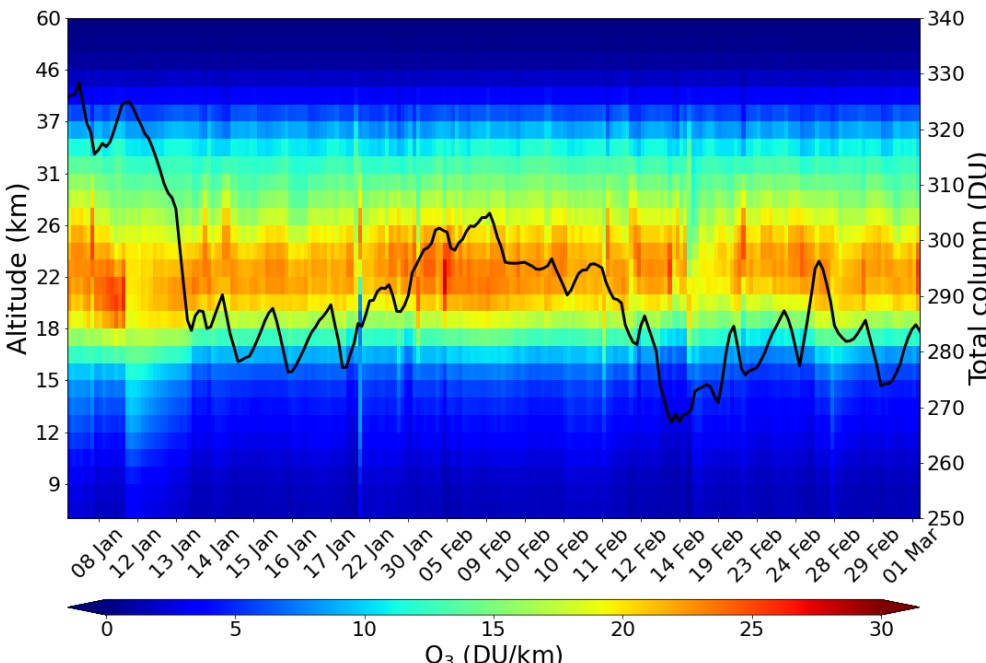

**Figure 7: Time series of the Lauder FTIR ozone partial column profiles for January and February 2020, with a 1-km vertical resolution. The black line (scale on the right) represents the corresponding total ozone column.**

**Table 1: Total ozone column and in-vortex stratospheric partial column (17-21 km), for selected days in January 2020 and their percent difference with respect to 1st-10th January average.**

| | Total Column (DU) 0-113km | Total column difference (%) 0-113km | Total column difference (DU) 0-113km | Partial column (DU) 17-21km | Partial column difference (%) 17-21km | Partial column difference (DU) 17-21km |
|---|---|---|---|---|---|---|
| **1:10-Jan** | 321.5 | 0.0 | 0.0 | 50.4 | 0.0 | 0.0 |
| **12-Jan** | 321.0 | 0.2 | 0.5 | 49.9 | 0.8 | 0.5 |
| **13-Jan** | 298.2 | 7.2 | 23.3 | 45.5 | 9.6 | 4.8 |
| **14-Jan** | 286.1 | 11.0 | 35.4 | 39.1 | 22.3 | 11.2 |
| **15-Jan** | 283.1 | 11.9 | 38.4 | 39.0 | 22.6 | 11.4 |

**Table 2: Same as Tab. 1 but for February.**

| | Total Column (DU) 0-113km | Total column difference (%) 0-113km | Total column difference (DU) 0-113km | Partial column (DU) 22-30km | Partial column difference (%) 22-30km | Partial column difference (DU) 22-30km |
|---|---|---|---|---|---|---|
| **5:10-Feb** | 300.6 | 0.0 | 0.0 | 165.1 | 0.0 | 0.0 |
| **15-Feb** | 269.8 | 10.2 | 30.8 | 135.4 | 18.0 | 29.7 |
| **17-Feb** | 260.9 | 13.2 | 39.7 | 129.9 | 21.3 | 35.2 |
| **18-Feb** | 272.3 | 9.4 | 28.3 | 137.1 | 16.9 | 28.0 |
| **19-Feb** | 272.2 | 9.4 | 28.4 | 144.1 | 12.7 | 21.0 |


By the time of the second overpass at Lauder, the high-aerosols/low-zone bubble was more compact and confined (see e.g. the evolution of the vertical shape of the ozone mini-hole in January in Fig. 2). To investigate further the vertical distribution of the ozone in the vortex, in the main vortex overpass phase in February, the individual FTIR observations data in February are shown in Figure 8. The period of observation starts the 15[th] of February (due to the lack of observations for the days immediately before), where an "S" shape can be observed in the vertical distribution of the ozone concentration. The ozone concentration is greatly reduced locally at the altitude of the vortex but seems unperturbed, with respect to monthly average,


at lower altitude than about 20 km, which underlines the very localised ozone reduction caused by this vortex after its separation from the main plume. Conversely, at altitudes immediately above the vortex a slight increase in the ozone concentration can be observed, similarly to what observed from satellite (Fig. 2). To investigate further the consistency of ground-based and

satellite observations of this particular vertical "S-shape" distribution, vertical OMPS-LP profiles in the core of the vortex shown in Figure 9. These show the mentioned "S-shape" quite consistently, with a localised ozone decrease at the vortex altitude and an increase right above the vortex altitude. Due to consistencies in ground-based and satellite-based observations, we tend to exclude inversion artifacts as causes of the observed "S-shape". The possible dynamical reasons of this vertical patterns are currently under investigation.


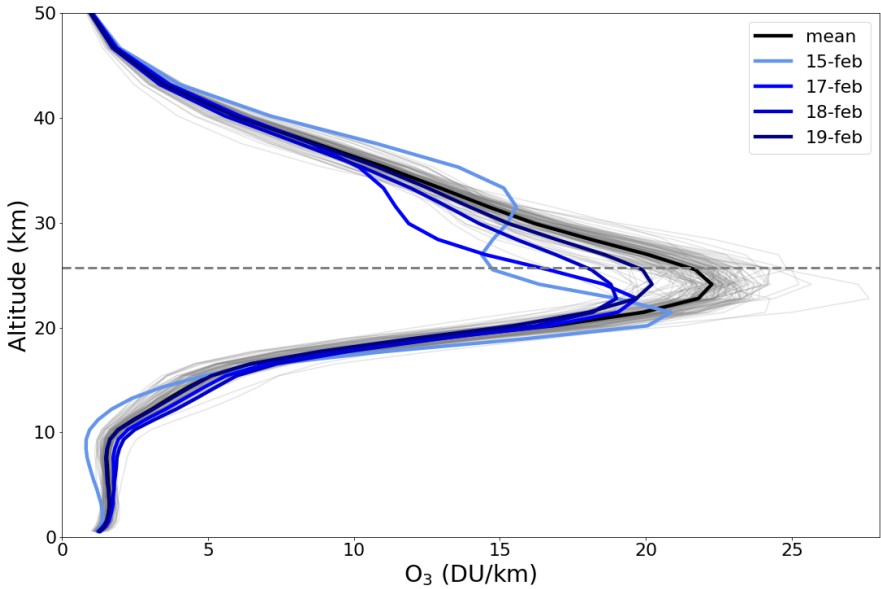

**Figure 8: Ozone concentration profiles for selected days in February 2020 from FTIR observations at the station in Lauder, New Zealand (lines in blue shades). The black line represents the mean ozone profile for the month of February, while the gray lines are**

**the profiles of each day of the month. The horizontal dashed line represents the altitude of the vortex for 15th February 2020.**

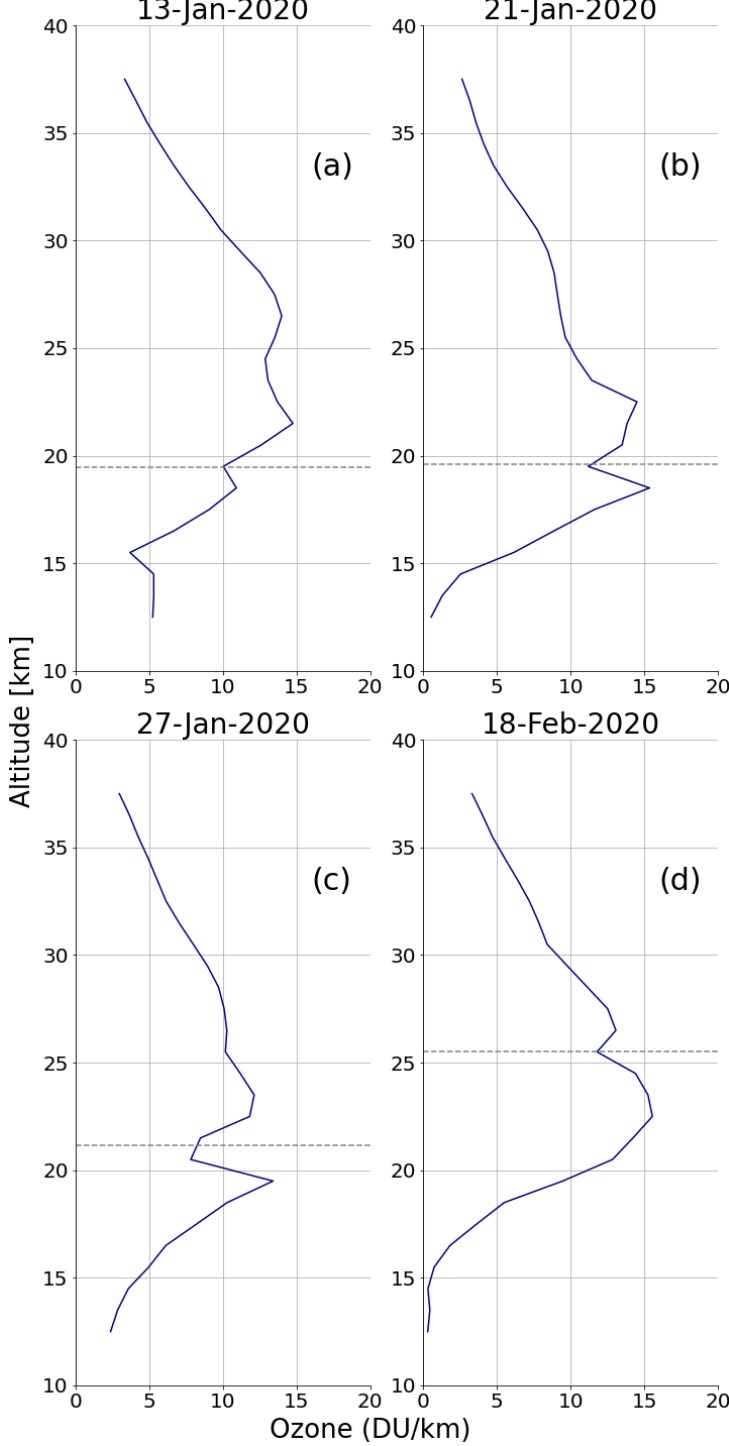

**Figure 9: Ozone vertical profiles from OMPS-LP on selected days in January and February 2020. The estimated altitude of the vortex is represented with dashed line.**

**Conclusion.**

The extreme fires in Australia in the 2019-2020 fire season were exceptional in many aspects. One remarkable feature is the formation of an anticyclonic confined smoke-containing vortex, generated by massive PyroCbs, that reached up to 35 km in altitude through diabatic radiative ascent associated with solar radiation absorption of smoke aerosols. This vortex was associated with an ozone-mini hole, due to rapid transport of ozone-poor tropospheric air masses to the ozone-rich stratosphere. Observations from satellite and ground-based instruments were used here, in combination with numerical reanalysis modelling

data, to study and quantify stratospheric ozone reductions of this mini-hole, and to track their horizontal and vertical dynamics, at the time scale of 1-2 months. In particular, with ad-hoc observations from the IASI infrared sounder and two different background-removal methods (a spatial and a temporal/climatological approach), we observed about 30-40% initial reduction of the in-vortex ozone column (chosen here as a progressively lofting 6-km layer, based on space LiDAR existing observations) with respect to background conditions, a few days after the main PyroCb injection. This reduction evolved following an

exponential decay law with an e-folding time of about 1 week and reached a longer-term equilibrium ozone anomaly of about 7%, possibly due to residual chemical depletion effects or an incomplete homogenization of the ozone concentration to the ambient stratospheric levels. This dynamically-generated ozone reduction is also observable in the total ozone column, with peak ~7% total ozone reduction observed by both the ultraviolet/visible TROPOMI satellite imagery and IASI. Based on our available data, we cannot conclude on the respective roles of chemical and dynamical origins of the ozone reduction. However,

it should be noted that the negative anomalies observed in this work do not exclude an absolute increase of the absolute ozone concentration in the vortex, as it is lofted to ozone-richer vertical atmospheric regions. In addition, the role of the dynamical isolation of the vortex in modulating the evolution of the ozone anomalies cannot be sorted out either, with our observations. Sellitto et al. (2023) suggest a limited mixing with the surrounding air in order to achieve the radiative heating necessary for the observed quick ascension in the stratosphere. This ozone reduction was also analysed from a favourable fixed location in

Lauder, New Zealand, thanks to ground-based FTIR observations. A clear transient reduction is observed from the ground as well, with reduction of 10 to 20% in a partial stratospheric column in January and February 2020. A particular vertical shape, with a localised reduction at the vortex altitude and a localised ozone increase at immediately higher altitudes is observed with both satellite data from the OMPS-LP limb profiler and the ground-based FTIR in Lauder. Further investigations are required to fully understand the dynamics of this unique event and the origins of this "S" shaped vertical patterns. The potential impacts

of this ozone mini-hole on the biosphere-harming ultraviolet radiation at the Earth's surface are also to be estimated. The findings from this study provide crucial insights into the effects of pyro-Cbs events and of large-scale wildfires on the atmospheric dynamics and composition.

**Financial support**

    This research has been supported by the Agence Nationale de la Recherche (grant no. 21-CE01-0007-01 (ASTuS)) and the

Centre National d'Etudes Spatiales (grant no. EXTRA-SAT).

## Author contributions

RB and PS designed the study. ME provided the IASI data. SB and TMN analysed the TROPOMI data. RB analysed the OMPS-LP data and calculated the in-vortex IASI columns. RB analysed the ground-based NDACC data. BL analysed the ECMWF-IFS data. All authors participated to the discussion of the results. RB, PS and BL wrote the manuscript and all authors contributed to its revision and editing.

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
