# Peer review of "Early evolution of the ozone mini-hole generated by the Australian bushfires 2019-2020 observed from satellite and ground-based instruments"

_EGUsphere, 2025_

## Author Comment (AC1)

**Reply to referees for the manuscript "Early evolution of the ozone mini-hole generated by the Australian bushfires 2019-2020 observed from satellite and ground-based instruments"**

Dear Editor, dear anonymous Referees,

Many thanks for your constructive criticism and the very useful comments. Thanks to your commitment to the review process, we have critically revised our manuscript. Please find more details and a point-by-point reply to the Referees' specific comments in the following (Referees' comments are in black and our replies in blue). We think that, thanks to your comments and suggestions, the present version of the manuscript is greatly improved with respect to the previous version.

Thank you very much,

Redha Belhadji on behalf of all co-authors

**Referee comments**

**RC2**

1) Page 3 L67: The sentence is not essential. Please delete it.

Done

2) Page 3 L80: The LISA algorithm for IASI ozone was used in this study. However, the IASI ozone algorithm also has the FORLI total ozone algorithm. What is the difference between the two methods? In addition, why is the LISA algorithm only used in this study?

The LISA ozone algorithm is totally different with respect to the FORLI algorithm. As mentioned in the text, the LISA ozone algorithm is described in the paper by Eremenko et al., 2008.

2) Section 2.2: Could you write the detailed specification of ECMWF-IFS? In the result section, the ECMWF-IFS specification (especial to the vertical resolution) is essential to understand the result.

Being based on hybrid pressure vertical coordinates, the actual vertical resolution of the ECMWF-IFS data is not fixed and hard to be generalized, even if it is safe to say that it is a few hundred meters in the stratosphere. In the Results section, the ECMWF-IFS profiles are of course degraded, in terms of the vertical resolution, to fit with the much scarcer vertical resolution of satellite observations.

3) Figure 2: This figure is too hard to see. Especially the right column of the figures are very difficult to check the detailed structure. Please correct it.

Done. Corrected the width of the circle that was wrong.

4) Figure 3: The author marked the 'star' in the figure to identify the center of the mini-hole. However, due to the 'star' mark, the intensity of the mini-hole is difficult to check. Please delete or change the color.

Done for all the plot with stars.

5) Page 8 L204-206: As mentioned before, the ECMWF-IFS vertical resolution is coarse. Also, the LISA ozone retrieval algorithm has its own vertical resolution. Could you clarify the vertical resolution of the LISA ozone result? In addition, how to calculate the exact '3km above and 3 km below' the center of the vortex? Could you explain the interpolation(?) method?

Please note that ECMWF-IFS vertical resolution is not coarse (see comment #2 above). For LISA ozone product, as mentioned in Sect. 2.1: "The LISA ozone retrieval scheme provides ozone profiles at 1-km vertical sampling and with 3.0-3.5 degrees of freedom for the whole atmospheric column and about 2.0 in the stratosphere (Dufour et al., 2012)." The in-vortex ozone column is obtained with linear interpolation based on the 1-km sampling vertical grid. Given this, determining ozone columns in -3km to +3km from the center (above and below the identified center of the vortex) is straightforward. Please also note that the fact that the IASI product has about 2.0 degrees of freedom in this vertical region, this 6-km wide partial ozone column has a much larger significance (e.g. in terms of smoothing error) than the ozone profile itself.

6) Page 8 L212: latitude and longitude ranges are too broad. Please explain the reasons why this coarse horizontal resolution is used in this study.

The vortex size is estimated to be around 1000 km in diameter. As a baseline, at 50° latitude, 70 km is approximately equal to 1° of latitude, so 1000 km corresponds to about 14°. This estimation also visually matches the observed size of the vortex.

7) Page 10 L247: Overall, I agreed that the aerosol plume affects the noise of ozone retrieval by IASI. However, this explanation needs a reference. Please include the related references that the IASI ozone retrieval is affected by the aerosol plume existence.

Yes, we agree with the Reviewer. We added the following reference:

Dufour, G., Eremenko, M., Beekmann, M., Cuesta, J., Foret, G., Fortems-Cheiney, A., Lachâtre, M., Lin, W., Liu, Y., Xu, X., and Zhang, Y.: Lower tropospheric ozone over the North China Plain: variability and trends revealed by IASI satellite observations for 2008–2016, Atmos. Chem. Phys., 18, 16439–16459, https://doi.org/10.5194/acp-18-16439-2018, 2018 (https://acp.copernicus.org/articles/18/16439/2018/).

8) Figure 5: Is this the temporal ozone anomaly? Please rephrase the exact physical variable (not simple O3 col).

Done.

9) Section 4.2: The explanation related to Tables 1 and 2 is not supported to the paragraph in L271-L286. Please rephrase it. In addition, I recommended that the difference be used to the 'absolute difference', not the 'relative difference'.

Tables 1 and 2 are introduced in the paragraph (at L275 of the original manuscript line numbering). We added the absolute to the already present relative difference in the two tables. We also limited the number of decimals to just one, in the revised manuscript.

10) Tables 1 and 2: What is the 0-113 km range?

It is the minimum/maximum altitude range of the FTIR observations, as provided by the data providers.

11) Figure 8: Could you check the partial degree of freedom of vertical distribution?

The FTIR ozone vertical profiles have about 4.5 degree of freedom, as estimated by e.g. Hassler et al., 2014 (https://amt.copernicus.org/articles/7/1395/2014/)

**RC1**

1) Abstract and throughout the article: I have a general problem with the following wording: ozone depletion! For me, ozone depletion is always linked to chemical (ozone decreasing) processes and not linked to a vertical ozone transport. I would avoid the word 'depletion' in this manuscript dealing with an ozone transport phenomenon.

Done

2) Page 2, line 42: Solomon et al. 2022 is not a good reference concerning wildfire smoke, it is more related to ozone depletion by smoke. Better references are Ohneiser et al (JGR, 2022) and all the Australian-smoke-related papers cited in this Ohneiser paper, and the recently published paper of Sakai et al. (JGR, 2025). Sakai et al. analyzed long-term lidar observations at Lauder, New Zealand. The time series covers the Australian bushfires.

Thank you for the suggestion. While we would like to keep the Solomon et al. reference – still quite general in the terms of the impacts wildfires on stratospheric chemistry, we also added here the references Ohneiser et al., 2022 (already cited elsewhere in the text). Unfortunately, we are not aware of the paper by Sakai et al. (2025) and cannot find it: in case important, please let us know the full title and we will have a look.

3) Page 2, line 56: Solomon et al. (2022) describes the impact of wildfire smoke on mid latitude stratospheric ozone, but there was even a more severe impact of the smoke on polar ozone as Ansmann et al. (ACP, 2022) highlighted. Should probably be mentioned.

Thank you, done

4) Page 3, line 64: vertical, not vortical.

Thank you, done

5) Page 5, line142: The smoke bubble was nicely observed over the tip of South America by Ohneiser et al. (ACP, 2022).

Thank you, done

6) Page 5, line 144: Many papers modelling the lofting of the smoke disk came to the conclusion that the BC content was just 2-3%.

Yes, this is discussed in one of the cited papers (Sellitto et al., 2023)

7) Page 6, line 166: The most intensive pyroCB phase was on 31 Dec 2019 and 1 Jan 2020, and not around 4-5 Jan. 2020 according to the report of Peterson et al. (npj, 2021) and summarized in the Ohneiser paper.

Thank you for the remark, we slightly modified the text accordingly.

8) Page 12, Figure 7: The vortex was transported to the east with the main westerly winds at heights below 20-25 km until 20-25 January, and after lofting to heights above about 25 km the vortex was transported to the west with the dominating easterly winds. Why is then the 'ozone hole' observed at heights around 22 km (in the west wind zone) over New Zealand on 17 February in Figure 7? The vortex was above 30 km height at that time…., in the easterly wind zone, and the ozone hole should be located around 30 km height.

The Figure 7 does not provide precise information about the height of the vortex; it is more focused on the ozone column. As stated in the following comment, on 17th February we observe the ozone depletion around 30 km altitude.

8) Page 14, Figure 8: The smoke bubble crossed New Zealand on 17 February, but not on 15 February, according to Figure 1. What is the reason for the pronounced local minimum in the ozone profile at about 26 km height on 15 February? The small ozone minimum in the ozone distribution around 30 km height observed on 17 February makes sense, and is linked to the wildfire smoke disk, but why is there a mini ozone hole on 15 February? Should be discussed!

The 15 February, the instrument at Lauder did not observe the core of the vortex. It observed the surrounding air masses that can be at a slightly lower or higher altitude than the core of the vortex.

---

## Author Response (AR2)

**Reply to referees for the manuscript "Early evolution of the ozone mini-hole generated by the Australian bushfires 2019-2020 observed from satellite and ground-based instruments", review stage two**

Dear Editor, dear anonymous Referee,

Many thanks for your constructive criticism and your very useful further comments. Based on these comments, we have revised our manuscript. Please find more details and a point-by-point reply to the Referee's specific comments in the following (Referee' comments are in black and our replies in blue). We think that, thanks to your comments and suggestions, the present version of the manuscript is improved with respect to the previous version.

Thank you very much,

Redha Belhadji on behalf of all co-authors

**Referee comments**

1) Abstract (lines 24–26): It may help to add that 30–40% ozone depletion is observed immediately after the event in the partial stratospheric column. This would emphasize the large values and help readers understand the contrast with the situation at the end of January and in the total column.

Done

2) Page 2, lines 50 and 53: Replace who with which.

Done

3) Page 2, line 52: Replace in his way with in its way.

Done

4) Page 2, line 42: As RC1 suggested, Solomon et al. (2022) is a strong reference describing chemical perturbations on chlorine and ozone. I suggest using it as a second reference. For the unprecedented magnitude of the event, however, the Sakai et al. (2025, JGR) paper may be more suitable: https://agupubs.onlinelibrary.wiley.com/doi/10.1029/2024JD041329?af=R

Done

5) Page 2, line 58: Replace where also performed with were also performed.

Done

6) Figures (Page 6, Fig. 1; Page 11, Fig. 6): The text refers to panels a and b, but these labels are not shown in the figures. Adding them would improve clarity.

Done

7) Page 6, line 159: use within instead of with. For example: "vertical evolution of the ozone mini-hole formed within the smoke vortex." Is the meaning intended to be: vertical evolution of the ozone mini-hole formed within the smoke vortex?

Done

8) Page 8, line 174: Replace 22th January with 22nd January.

Done